# Event-Based Vision Application on Autonomous Unmanned Aerial Vehicle: A Systematic Review of Prospects and Challenges

**DOI:** 10.3390/s26010081

**Published:** 2025-12-22

**Authors:** Ibrahim Akanbi, Michael Ayomoh

**Affiliations:** Department of Industrial and Systems Engineering, University of Pretoria, Pretoria 0002, South Africa

**Keywords:** event camera, dynamic vision sensor, unmanned aerial vehicles (UAVs), neuromorphic sensor, obstacle avoidance, navigation, path planning

## Abstract

Event camera vision systems have recently been gaining traction as swift and agile sensing devices in the field of unmanned aerial vehicles (UAVs). Despite their inherent superior capabilities covering high dynamic range, microsecond-level temporary resolution, and robustness to motion distortion which allow them to capture fast and subtle scene changes that conventional frame-based cameras often miss, their utilization has yet to be widespread. This is due to challenges like insufficient real-world validation, unstandardized simulation platforms, limited hardware integration and a lack of ground truth datasets. This systematic review paper presents an investigation that seeks to explore the dynamic vision sensor christened event camera and its integration to (UAVs). The review synthesized peer-reviewed articles between 2015 and 2025 across five thematic domains, datasets, simulation tools, algorithmic paradigms, application areas and future directions, using the Scopus and Web of Science databases. This review reveals that event cameras outperformed traditional frame-based systems in terms of latency and robustness to motion blur and lighting conditions, enabling reactive and precise UAV control. However, challenges remain in standardizing evaluation metrics, improving hardware integration, and expanding annotated datasets, which are vital for adopting event cameras as reliable components in autonomous UAV systems.

## 1. Introduction

This section and the following sub-sections ranging from 1.1 to 1.3 present introductory technical information covering the study’s background, introducing and articulating the concept of event camera vision system, their operating principles and the advantages of event cameras over conventional frame-based cameras for fast autonomous UAV operation. The discussion covers how event cameras asynchronously capture brightness changes at individual pixels, enabling high temporal resolution and low latency. Additionally, the sections highlight key features such as high dynamic range and reduced data redundancy, which make event cameras well-suited for fast and challenging visual environments. The background also touches on the challenges of processing event-based data and the need for specialized algorithms to fully leverage their unique output characteristics.

### 1.1. Background of the Study

Event cameras, sometimes referred to as dynamic vision sensors (DVSs), are a paradigm shift in visual sensing since they do not focus on taking full-frame pictures but rather on catching scene changes [1]. They represent a paradigm shift in the collection of visual data since they are asynchronous sensors. This is because, as opposed to using a clock that is unrelated to the image being watched, they sample light based on the dynamics of the scene [1]. In contrast to conventional cameras, whose pixels have a common exposure time, event-based cameras function asynchronously at the pixel level with microsecond-scale resolution [2]. This is especially helpful in situations with fast action, as typical cameras might blur motion or need unreasonably high frame rates to capture details [3]. This speed is essential for real-time UAV operations such as visual SLAM, obstacle avoidance, odometry, and collision prevention under low-light conditions. 

UAVs, commonly known as drones or unmanned aircraft systems (UASs), have experienced popular adoption in various sectors which include but are not limited to entertainment, military, precision agriculture, smart city systems, wildlife conservation and monitoring, logistics, and delivery services, amongst others, highlighting the critical need for a sophisticated and advanced aerobotics navigation system with swift dynamic capabilities, covering fast-moving object state space coordinate tracking, space collision prevention, and obstacle avoidance. Originally, UAVs were mostly used for military purposes, where they were very important for reconnaissance, surveillance, and targeted operations [4], but these intelligent agents have proven significant in the advancement of smart city transit systems [5], precision agriculture [6], the support of search and rescue operations [7], shipping and delivery such as with Amazon Air [8], aerial photography [9], wildlife monitoring and conservation [10], and entertainment [11], among several others, and they are also expected to be a big part of making cities more connected and automated as part of Industry 4.0 and the push for smart cities. Despite the widespread adoption of these UAVs, there have been several reports of crashes due to collisions with static and dynamic obstacles as highlighted by [12] and as shown in Figure 1, which shows analysis of 60 UAV accident reports, identifying that design flaws and pilot response issues were the key causative factors. These urban environments, characterized by high-rise buildings, utility poles, and other obstacles, underscore the necessity of sophisticated avoidance techniques to mitigate potential collision risks. Likewise, ref. [13] highlights how adverse weather, like heavy rain, compromises UAV detection accuracy by obscuring camera vision and degrading sensor performance. UAVs can perform repetitive tasks more efficiently, but only if they can navigate accurately. This requires them to process information and make decisions quickly, as well as to perceive their environment with high speed and precision. Achieving this level of autonomous navigation is crucial for UAVs to operate effectively, especially in dynamic environments where rapid responses and adaptability are essential [14].

In the last few years, there has been tremendous work by various researchers in using event camera vision systems for fast autonomous navigation of UAVs in a dynamic environment. This vision system offers a paradigm shift by capturing changes in brightness asynchronously, providing high temporal resolution, low latency, low power consumption, and high dynamic range, and has been widely adopted not just for autonomous navigation in UAVs but in the entire field of computer vision ([15,16,17,18], etc.) Leveraging event camera vision systems for dynamic obstacle avoidance in UAVs opens up numerous practical applications, including aerial imaging, last-mile delivery, and urban air mobility markets which are experiencing rapid growth and are worth billions of dollars, and they are forecasted to grow to USD 132.36 billion by 2035 [19]. This capability is especially significant due to the safety concerns associated with operating aerial vehicles above crowds, as recent incidents have highlighted the risks posed by drones colliding with birds or objects thrown at quadrotors during public events. By reducing the temporal latency between perception and action, this technology helps prevent collisions and non-negligible risk factors in urban environments as well as severe hardware failure which could lead to losses [20]. These characteristics make them perfect for robotics and computer vision applications where conventional cameras are ineffective, like situations requiring high dynamic range or speed [21].

Autonomous drones without event cameras react within tens of milliseconds, which falls short for swift navigation in complex, dynamic environments. To safely avoid collision with fast moving objects, drones need sensors and algorithms with minimal latency [22]. Similarly, Ref. [23] highlights the necessity of low latency for navigating unmanned aerial vehicles around dynamic obstacles. Event cameras stand out in these contexts due to their high dynamic range. For instance, Ref. [24] proposed an entirely asynchronous method for monitoring intruders using unmanned aircraft systems (UASs), leveraging event cameras’ unique properties. Compared to conventional cameras, event cameras offer significant advantages such as high temporal resolution (in milliseconds), an exceptionally high dynamic range (140 dB versus the typical 60 dB), low power consumption, and high pixel bandwidth (in kHz), which minimizes motion blur. Consequently, event cameras show strong potential for robotics and computer vision in scenarios where traditional cameras may fall short. They also produce a sparser and lighter data output, making processing more efficient [21,22].

The integration of event-based vision in UAV systems represents a critical juncture in the evolution of aerial autonomy. While numerous individual studies have explored aspects of this integration, the existing body of knowledge in UAVs remains fragmented. Researchers face a lack of consolidated information regarding the current state of event camera usage in UAVs, especially in areas such as publicly available datasets with ground truth, simulation environments, algorithmic developments, and real-world applications. Despite several notable similar reviews like [21,25], none of them have been able to focus on UAVs, which are a technology that is receiving global attention and requires a critical approach for automation. This fragmentation poses a barrier for newcomers in the field of UAVs who should leverage the features of this camera over standard cameras.

This systematic literature review (SLR) seeks to bridge this gap by synthesizing recent advancements, identifying core limitations, and uncovering future possibilities for event cameras in UAV applications. It highlights the critical need for event cameras for fast autonomous sensing in UAVs to enable rapid responses to dynamic and complex environments. As shown in Figure 2, this review is divided into five sections. In Section 1, we discuss the background of UAVs and the need to leverage event cameras for fast autonomous navigation. Section 2 discusses how we used the common PRISMA approach to organize articles for the systematic literature review. Then in Section 3 we discuss the results of our findings, which included various models and algorithms researchers have used in several UAV applications using this camera. Here we categorize the methods into geometric approach, learning-based, neuromorphic, and hybrid approaches. It also details event camera applications in UAVs, also covering the available datasets and simulators. Section 4 provides a descriptive analysis of the review results. It emphasizes the advantages and relevance of event cameras for UAV applications, elaborating their edge over frame-based cameras in autonomous aerial systems. And finally, in Section 5, the conclusion reiterates the potential of event camera sensor to revolutionize UAV performance in dynamic environments despite current obstacles such as lack of standardize evaluation, inadequate real-word validation and immature simulation platform. 

The review is guided by five interrelated objectives:(i)To examine existing algorithms and techniques spanning geometric approaches, learning-based approaches, neuromorphic computing, and hybrid strategies for processing event data in UAV settings. Understanding how these algorithms outperform or fall short compared to traditional vision pipelines is central to validating the potential of event cameras.(ii)To explore the diverse real-world applications of event cameras in UAVs, such as obstacle avoidance, SLAM, object tracking, infrastructure inspection, and GPS-denied navigation. This review highlights both the demonstrated benefits and operational challenges faced in field deployment.(iii)To catalog and critically assess publicly available event camera datasets relevant to UAVs, including their quality, scope, and existing limitations. A well-curated dataset is foundational for algorithm development and benchmarking.(iv)Identify and evaluate open-source simulation tools that support event camera modeling and their integration into UAV environments. Simulators play a vital role in reducing experimental costs and enabling reproducible research.(v)To project the future potential of event cameras in UAV systems, including the feasibility of replacing standard cameras entirely, emerging research trends, hardware innovations, and prospective areas for interdisciplinary collaboration.

By organizing the literature according to these five thematic pillars, this review offers a structured resource for scholars, engineers, and practitioners in robotics, computer vision, and autonomous systems working on UAV navigation and perception. Furthermore, it identifies unresolved challenges, benchmarks current progress, and proposes directions for future work, aiming to accelerate innovation and practical adoption of event-based vision in autonomous aerial systems.

### 1.2. Basic Principles of an Event Camera

Event cameras are bio-inspired sensors that differ from conventional cameras due to their asynchronous way of measuring per-pixel brightness changes compared to the former, which captures images at a fixed rate [21]. Every pixel in an event camera continually and independently tracks changes in intensity. A pixel that notices a notable shift in light intensity, either increasing or decreasing, creates an “event” that contains its location, the polarity of the change which is brightening or darkening, and an exact timestamp [21,26]. The fundamental idea underlying event cameras is the asynchronous recognition of scene changes, enabling them to function with great temporal resolution, frequently in the microsecond range.

The capacity of event cameras to function in difficult lighting settings is another important benefit. Conventional cameras would find it difficult to simultaneously capture bright and dark areas in scenes with great dynamic ranges, whereas event cameras are simply sensitive to changes in intensity. Because only pixels that undergo a change are captured, the data produced by event cameras is also sparse and compact, requiring less data bandwidth and processing that is more effective [27]. Because of these characteristics, event cameras are perfect for robotics, autonomous driving, and surveillance applications where quick and low-latency vision is essential. Nevertheless, the asynchronous character of the data presents difficulties for conventional computer vision algorithms, requiring the creation of fresh processing methods customized for the data format of event cameras [23,27].

Conventional image sensors and event-based cameras function very differently. Event-based cameras offer great temporal resolution and little motion blur, while traditional cameras collect images at fixed frame rates. Event-based cameras detect changes in pixel intensity asynchronously. This makes event cameras perfect for situations where standard sensors frequently falter, such as high-speed or low-light conditions. Furthermore, event cameras interpret sparse data and use less power, which improves efficiency in real-time applications like UAV navigation [23]. Traditional cameras, on the other hand, work better in static contexts (such as object recognition jobs) when full-frame information is essential. Though integrating event cameras with traditional computer vision processing algorithms is still a problem, they perform best in dynamic environments where only little changes are noticeable [27].

### 1.3. Types of Event Cameras

This subsection presents the different categories of event cameras. The itemization herein spans across five different types, as highlighted in Table 1. Their detailed mode of operation and identified gaps were also presented.

## 2. Materials and Methods

This review followed the Preferred Reporting Items for Systematic Reviews and Meta-Analyses (PRISMA) approach [35] as shown in Figure 1. Five main research questions are addressed in this study: which open-source simulation tools facilitate the integration of event cameras into UAV systems; which publicly available event camera datasets are there for UAV applications and their limitations; which major models or algorithms have been developed for event-based UAV perception and how well do they perform in comparison to standard vision systems; which UAV applications have successfully deployed event cameras and what challenges have arisen; and which emerging future directions and innovations for event cameras in UAV applications are being explored. Consistent with common practices in the engineering literature, the protocol was developed, reviewed, and archived internally without registration.

### 2.1. Search Terms

We used extensive database coverage across two major academic repositories, Scopus and Web of Science. These interdisciplinary engineering databases were chosen for their comprehensive coverage of peer-reviewed publications. Boolean operators and wildcards were used strategically with phrases like “Event camera?” OR “Event-based camera?” OR “dynamic vision sensor?” OR “DVS” AND (“UAV” OR “drone?” OR “unmanned aerial vehicle?”)

### 2.2. Search Strategy and Criteria

Inclusion and Exclusion Criteria:Inclusion Criteria: Peer-reviewed journal articles and conference proceedings that directly applied event cameras in UAV contexts, with empirical evaluation of systems, algorithms, or datasets related to UAV navigation, perception, tracking, SLAM, or object recognition.Exclusion Criteria: Publications prior to 2015, non-English studies, duplicate publications, secondary summaries, and research focusing solely on hardware design or biological vision systems without any application to UAV robotics.

Screening and Selection Process:Identification: An initial total of 245 records were identified from Scopus (*n* = 195) and Web of Science (*n* = 50)Removal of Redundancies: Duplicate records (*n* = 38) and non-English records (*n* = 8) were removed.Screening: The remaining 199 records were screened based on titles and abstracts. This screening phase excluded 18 records.Retrieval and Eligibility Assessment: Reports sought for retrieval totaled 181, with 30 not retrieved. The remaining 151 reports were assessed for eligibility. During this assessment, 22 reports were excluded because their focus was solely on hardware design or biological vision systems without applications in UAV robotics.Final Selection: A total of 129 relevant papers were ultimately included in the review. These selection processes are indicated in Table 2.

### 2.3. Data Extraction

Data extraction was performed using a custom structured matrix designed to capture the bibliographic information, methodological characteristics, and technical categories of each paper. Initially, all identified records downloaded from Scopus and Web of Science were exported in RIS format and imported into Mendelay Reference Manager (version 2.138.0) where duplicates were removed automatically. The cleaned reference list was then exported into a csv file for systematic extraction. This structure matrix recorded key bibliographic data (authors, title, publication year, document type, and source), study specifications (selected, dataset, algorithm or model method, and fusion), and technical details which included the event camera model used.

The supported tools included Python (version 3.12.0) with Pandas (version 2.2.3) for quantitative summaries and exploratory data visualization, Vosviewer (version 1.6.20) for clustering, Microsoft Excel for data analysis and cross-tabulation, and Mendeley for source organization and citation management. Five standardized criteria were used to evaluate each study for quality assessment: reproducibility through the availability of source code, datasets, or clear implementation details; methodological rigor through valid experimental design and evaluation procedures; innovation and contribution through novel techniques or applications for event-based UAV perception; empirical validation with quantitative results and benchmark comparisons; and clarity of objectives regarding event camera-based UAV research goals. A binary system was used to score the studies; papers that satisfied at least four of the five requirements were given priority as high-quality contributions. The lead reviewer carried out the quality assessment independently, using secondary verification to ensure consistency and reduce bias.

The flowchart as indicated in Figure 3 illustrates the systematic selection process of relevant event-based camera studies in UAVs, highlighting the filtering of the initial search results to the 129 studies that confirmed the review’s conclusions.

Using Vosviewer software [36], Figure 4 demonstrates the interdisciplinary nature of event-based UAV research, highlighting strong connections between neuromorphic sensing, autonomous navigation, and real-time visual processing, which form the foundation of current event-driven aerial navigation development.

## 3. Results

### 3.1. Review of Past Survey of Event Cameras in UAV Applications

The field of event-based vision and its application in autonomous unmanned aerial vehicles has been rapidly evolving, with various reviews and surveys addressing different aspects of this technology. Early work by [21] covers the event camera’s principles, algorithms, hardware, and applications across various tasks, but it is limited to a general robotics and computer vision context, and it lacks an in-depth review of event camera-specific SLAM and real-time state estimation methods tailored specifically to UAVs. Similarly, Ref. [37] reviewed vision-based navigation system, covering different sensors such as stereo and RGB devices, LIDAR camera hybrids, event cameras, and infrared systems, highlighting SLAM algorithms and control strategies across robotic domains. However, it lacked focused analysis of event-based SLAM and real-time UAV applications. Additionally, Ref. [15] detailed several publicly available event camera datasets relative to automotive in-cabin and out-of-cabin scenarios, sensor fusion, optical flow, and depth estimation, but it does not address or provide detailed discussion on the application of event cameras in autonomous UAVs. Thus, this survey underlines the need for targeted research and comprehensive systematic reviews that address event-based perception, SLAM, and control methods specific to UAV platforms to close these gaps and fully exploit this sensor’s potential in aerial autonomy.

### 3.2. Models and Algorithms

Event processing data for events in UAVs is an important aspect because the use of algorithms is considered in the development of interpreting applications associated with event data in UAV. As a result of this, the asynchronous and inimitable landscape of event streams, algorithmic approaches differ to a large extent from those which are adopted in frame-based visualizations. Owing to this, grouping these algorithms into four distinct categories can be considered, such as the following: model-based approach, learning-based methods, neuro-morphic approach of computing, and hybrid sensor integration methods.

#### 3.2.1. Geometric Approach

The geometric approach of processing is the foundational category of methods used in ego-motion compensation, dense estimation, and optical flow for UAVs equipped with event cameras. This is based on the principles of projective geometry and rigid body transformations [38]. Studies have identified a method for a real-time optical flow using a DVS on a miniature embedded system that is suitable for autonomous flying robots. The local motion at each pixel is modeled as a linear combination of the three global motion components, pan, tilt, and yaw rotations, which are represented by vectors $v_{x}$ and $v_{y}$, where the local flow v(x,y) at each pixel is expressed as follows:(1)$$v\left(x,y\right)=\;\propto v_{x}+\;\beta v_{y}+\;\gamma v_{z}\;$$
where α, β, γ are the coefficients representing pan, tilt, and yaw, respectively.

Event-based visual odometry is identified as part of the major dominant and first recognized technique which was earlier introduced by [29], and it was found that the performance features of this approach can be used to track and estimate camera motions that do not have frames and also to attain a rotation error of about 0.8° and a translation error close to 2%, with computational efficiency which is appropriate for onboard UAV processing. Conversely, Ref. [39] argued against this assertion in comparison to a visual odometry technique with low latency which eventually ensures that delays are minimized.

Contrast maximization, on the other hand, was introduced by [40] and is known to take an entirely different route by optimizing alignment through motion-compensated contrast development in the event stream. Meanwhile, it is known to be powerful in scenes that are static, and its assumption of inflexibility exposes it to vulnerabilities from unique interference.

The dynamic vision sensors are bio-inspired with sensors that record intensity changes rather than intensity of images from pixel-wise captures.

Given an event $e_{k}\dot{=}\left(x_{k},\;t_{k},\;p_{k}\right)$ that was triggered by a logarithm intensity $L\left(x_{k},t_{k}\right)$ at a given pixel that is more than the contrast intensity C>0, the logarithm intensity can be modeled by [28] as follows:(2)$$L\left(x_{k},t_{k}\right)-L\left(x_{k},t_{k},{∆t}_{k}\right)=p_{k}C,\;$$
where $x_{k}={\left(x_{k},y_{k}\right)}^{T}\;$ is the spatio-temporal coordinates $\;in\;t_{k}$ (with μS resolution), polarity $p_{k}\in \{+1,-1\}$ is the sign of the change in intensity, and $t_{k}-∆t_{k}$ is the time difference between the pixels.

Gallego et al. (2018) [40] modeled the contrast maximalization with a mathematical framework from a dynamic vision sensor. Given a set of events $\zeta ={\left\{e_{k}\right\}}_{k=1}^{N_{e}}$,(3)$$e_{k}\dot{=}\left(x_{k},\;t_{k},\;p_{k}\right)\;\to \;e_{k}\dot{=}\left(x_{k},\;t_{ref},\;p_{k}\right)\;$$

According to their work, the motion model W results in a set of warped events $\zeta^{\prime }={\left\{{e_{k}}^{\prime }\right\}}_{k=1}^{N_{e}}$. This results in a warp given as follows:(4)$$x_{k}^{\prime }=W\left(x_{k},\;t_{k},\theta \right)\;$$

The warp transport events along the motion point trajectory θ until a time $t_{ref}$ has been reached. An objective function called the image of warped events is modeled to measure the alignment of the warped events as follows:(5)$$I\left(x;\theta \right)=\;\sum_{k=1}^{N_{e}}b_{k}\delta \left(x-x_{k}^{\prime }\left(\theta \right)\right)\;$$
where x is the pixel and sums the values for the warped events that fall within its range of $b_{k}=\left[\;p_{k}\;1\right],\;$ with the former given the value if polarity was used and 1 if it was not used.

Continuous-time trajectory estimation was proposed by [41], which is an identified motion model connected with a continuous function instead of discrete poses, which aligns better with the asynchronous nature of event data. Furthermore, Ref. [42] introduced the EVO, which is an identified 6-DOF system of mapping and parallel tracking used to process events in a timely manner; meanwhile, it was found that the performance of the EVO lacks low-texture settings. Ref [29], in their work, explored how standard cameras use fixed frame rates to send full frames and how event cameras use their independent pixels to continuously fire intensity changes in the image plane. For the given intensity I, the sensor generates an event at the point $u{\left(x,y\right)}^{T}$ with a logarithm function given as follows:(6)$$|∆\log\left(I\right)|\approx \;\left|-\left\{\nabla \log\left(I\right),\dot{u}∆t\;\right\}\right|>C\;$$
where ∇ calculates the special gradient in the motion field $\;\dot{u\;}$ over a time frame ∆t. These events are recorded with a timestamp and asynchronously transmitted due to advanced digital technology that works behind the scenes. The events of the cameras usually form tuples of $e_{k}=\;\left(x_{k},\;t_{k},\;p_{k}\right)$, where $\left(x_{k},\;t_{k}\right)$ forms the coordinates of the event, polarity is $p_{k}$, and the timestamp is $t_{k}$.

In practice the Delta function in the IWE is replaced with a smooth approximation like the Gaussian, such that(7)$$\delta \approx N\left(x;\mu ,\epsilon^{2}\right),\;\;with\;\epsilon =1\;pixel\;$$

The objective function of the IWE model is(8)$$G\left(\theta \right)=Var\;I\left(x;\theta \right)\;\dot{=}\;\frac{1}{\left|\Omega \right|}\;\underset{\Omega }{\int }\dot{=}{\left(I\left(x;\theta \right)-\mu_{I}\right)}^{2}\;dx,\;$$
where $u_{I}$ is the mean and Ω is the image domain.(9)$$\mu_{I}\;\dot{=}\;\frac{1}{\left|\Omega \right|}\;\int_{\Omega }\dot{=}\left(I\left(x;\theta \right)\right)\;dx,\;$$

Hence, the optimization algorithm that optimizes the contrast framework is given as follows:(10)$$\theta^{*}=\arg\underset{\theta }{\max}\;G\left(\theta \right)\;$$

Therefore, the model-based approach is found to maintain an attractive computational suitability and simplicity for UAVs which are resource-constrained; meanwhile, their limitations are based on the sparsity of scene, sensitivity to noise, and element dynamism. Table 3 provides a comprehensive list of annotations used in this geometric approach. And contributions using the geometric approach are summarized in Table 4.

#### 3.2.2. Learning-Based Methods

Deep learning techniques have been developed to overcome many of the drawbacks of model approaches, especially when it comes to managing dynamic, complicated scenes.

E2VID was one of the earliest models to translate event streams into standard frames for CNN processing [48], and the main temporal benefits of event data were jeopardized, even if this allowed for high-quality image reconstruction. When applied to autonomous driving, ref. [49] demonstrated that this technique worked well for simple navigation.

EV-FlowNet used self-supervised optical flow estimation; Ref. [50] preserved the event structure, attaining exceptional accuracy (0.32 average endpoint error) and resilience in demanding settings.

Dynamic tracking based on events has also advanced and strong object detection techniques for harsh illumination situations were created by Mitrokhin et al. [1,51], and they were extended using the EV-IMO dataset [51]. Traditional Transformer-based approaches, such as the Vision Transformer (ViT), face the challenge of high computational complexity due to excessively long tokens [52]; however, Cross-Deformable-Attention (CDA) modules have been designed to significantly reduce computational complexity. Table 5 is the summary of the relevant contribution using leaning-based approaches.

#### 3.2.3. Neuromorphic Computing Approach

Neuro-morphic approaches seek to maintain the biological analogies of event data because of their spike-like characteristics. These techniques are particularly applicable to UAVs with limited power. Ref. [60], a study on using drones for civil-infrastructure inspection, demonstrates that pairing event cameras with SNNs can drastically cut energy while preserving accuracy. It shows that SNNs on Loihi are 65–135× more energy-efficient than ANNs on a modern accelerator, with only a 6–10% drop in defect-classification metrics and better robustness under extreme lighting, but it remains a classification-only pipeline that assumes a conventional flight controller and does not address full perception–action loops or dense scene understanding. Ref. [61] then fills that gap on the perception side by moving from image-level labels to pixelwise semantic maps. It proposes a fully spiking, U-shaped encoder–decoder architecture for event data that uses PLIF neurons, avoids batch normalization, cuts parameters by about 1.6× relative to the closest spiking baseline, and still improves MIoU by around 5.6 percentage points, yet its focus is on the offline driving DDD17 dataset and its authors consider full deployment on neuromorphic chips like Loihi or SpiNNaker as future work rather than demonstrating an embedded, closed-loop system.

Ref. [62] moves explicitly into closed-loop control, but only for the mid-level behavior of obstacle avoidance in a software simulation experiment with XTDrone. The DNN detection method replaces heavy SNN detectors with Spiking-YOLO, which uses 7.9M neurons with a 3072-neuron component, and couples this to Kalman and Bayesian predictors plus confidence-interval logic to produce safe velocity commands that can avoid obstacles with up to 8 m/s relative speed within 0.2 s, even under missed detections. However, it still assumes a conventional flight controller beneath it and leaves actual deployment on neuromorphic processors, integration with richer semantic perception, and truly end-to-end neuromorphic control as explicit future directions.

Ref. [63] presented a fully neuromorphic vision by running an entire perception-to-actuation pipeline on spiking networks fed by an event camera and driving a flying drone directly. It accepts raw ego-motion and scene events, learns low-level control via supervised learning in a simulator, and then flies a real quadrotor at 200 Hz update rate on neuromorphic hardware, using only about 0.94 W for inference plus roughly 7–12 mW for on-board learning, while robustly performing hovering, landing, sideways maneuvers, and turning. A promising avenue for this drone is making sensing, processing and actuation fully neuromorphic, but this is currently limited by available neuromorphic hardware and I/O, so full onboard neuromorphic pipelines remain a hardware-driven future goal. In Table 6, we summarized all the relevant contributions using the neuromorphic approach.

#### 3.2.4. Hybrid Sensor Integration Methods

The drawbacks of single-sensor systems are addressed by hybrid techniques, which combine event data with various sensor modalities. Ultimate SLAM by [39] integrates IMU, frame, and event data to provide reliable SLAM under high-speed, HDR circumstances. Stereo event processing and sensor fusion are supported by the MVSEC dataset [29]. Ref. [75] improved on this model when they enhanced it with a range sensor and called the model REVIO. This model outperforms existing methods on the event camera dataset, reducing position error by up to 80% in high-speed scenes and achieving better accuracy and efficiency compared to [39] and VINS-Mono in dynamic environments. All the hybrid-based approaches are summarized in Table 7.

### 3.3. Application Benefits of Event Camera Vision Systems in UAVs

The use of event camera vision systems in unmanned aerial vehicles (UAVs) has made it possible to use them in a variety of fields where traditional frame-based vision systems are often ineffective. The high temporal resolution, low latency, and durability of event-based cameras in dynamic and low-light-environment conditions frequently encountered in airborne operations motivate their use in UAV systems. Consequently, a number of creative use cases have surfaced in both experimental and research deployments, and this section summarizes important application areas found in the literature, demonstrates how event-based vision improves performance in each situation, and considers lingering restrictions and integration difficulties.

#### 3.3.1. Visual SLAM and Odometry

Visual odometry and event-based SLAM are two of the most studied topics for UAV navigation with event camera vision systems. The use of event camera vision systems in UAVs for visual odometry (VO) and simultaneous localization and mapping (SLAM) is among the oldest and most actively researched uses of these cameras. EVO [42] and Ultimate SLAM [39] are two examples of systems that show how event streams may be used for precise 6-DOF pose tracking in high-speed motion and in settings where motion blur or changing lighting would cause classic frame-based SLAM to fail. In fast-moving aerial situations, motion blur and latency are the limitations of traditional frame-based SLAM systems. On the other hand, by accurate temporal sampling, event cameras allow for continuous time pose estimation.

Ref. [29] showed that event-based visual odometry could accurately estimate UAV motion latency-free [39] expanded on this work with the Ultimate SLAM framework, which achieves robust SLAM in high-dynamic-range (HDR) situations by combining event data, frames, and inertial measurements.

These systems performance can degrade in low-texture settings or during aggressive maneuvers, despite their potential in organized environments. This suggests that more algorithmic robustness and sensor fusion techniques are required.

#### 3.3.2. Obstacle Avoidance and Collison Detection

Event cameras’ low latency and resistance to motion blur have allowed them to perform exceptionally well in reactive obstacle avoidance and high-speed navigation. For high-speed obstacle avoidance in UAVs, event cameras are perfect because of their lightning-fast response and lack of motion blur. While event cameras may detect changes in the visual field in microseconds, traditional vision-based systems may not be able to detect fast-moving impediments in dynamic situations. Ref. [53] have tested and proposed reactive systems that can identify and avoid moving objects in milliseconds. Quadrotors can avoid fast dynamic obstacles using event cameras with 3.5ms latency, as demonstrated by [22] and event-based moving object detection frameworks were created by [1] and they showed dependable segmentation in challenging motion and lighting scenarios. For ornithopter UAVs, Ref. [24] implemented a biologically inspired sense-and-avoid system that uses asynchronous event data to enable evasive maneuvers with response times of less than a millisecond. These investigations highlight the effectiveness of event-based systems in situations that call for prompt decision-making, like autonomous defense applications, drone racing, and spying.

However, there are still unresolved issues with filtering noisy activations and adjusting thresholds for event triggering, especially in cluttered, multi-object environments.

#### 3.3.3. GPS-Denied Navigation and Terrain Relative Flight

In environments where GPS is not available, including tunnels, urban canyons, woodlands, or indoor spaces, UAVs must rely on vision-based navigation. An appealing substitute for conventional visual-inertial systems, event cameras allow for terrain-relative navigation that swiftly adjusts to changing conditions. For localization and map-less landing, some solutions have integrated downward-facing sensors with event cameras.

To accomplish low-power, precise localization in restricted regions, Ref. [81] introduced NeuroSLAM, a mixed-signal neuro-morphic SLAM system that takes advantage of event camera data.

Although promising, these techniques still need a lot of sensor fusion with depth sensors and inertial data to guarantee stability over extended missions.

#### 3.3.4. Infrastructure Inspection and Anomaly Detection

Event cameras have been mounted to unmanned aerial vehicles (UAVs) in the fields of civil engineering and smart infrastructure to perform fine-grained inspection jobs including detecting building flaws or bridge cracks. Visual systems that can function in challenging or fluctuating lighting circumstances are necessary for UAV-based inspection of vital infrastructure, such as buildings, bridges, and power lines. Event cameras are ideal for capturing fine details in areas that are overexposed or shaded because of their HDR capabilities.

The ev-CIVIL dataset was created by [82] especially for infrastructure assessment with event cameras installed on unmanned aerial vehicles. Their work showed how to successfully identify civil structural flaws in highly contrasted illumination. These applications are especially pertinent to automated maintenance workflows and smart city monitoring.

Notwithstanding these benefits, the area does not yet have large-scale annotated datasets or established criteria for comparing event-based flaw detection.

#### 3.3.5. Object and Human Tracking in Dynamic Scenes

In situations where there are numerous moving agents, such as in search and rescue operations or disaster response areas, event cameras have demonstrated the ability to detect and track humans or vehicles [79]. UAVs have employed event camera vision systems to perform aggressive flight maneuvers, such as making sharp turns, avoiding swift objects, and trajectory prediction. Refs. [76,83], in their event camera vision system, demonstrate dynamic tracking.

In order to enhance tracking performance in extremely dynamic or dimly lit conditions, ref. [84] suggested a hybrid human identification framework for UAVs that combines traditional vision with event streams. They demonstrated enhanced resilience to background motion and occlusion with their multimodal curriculum learning strategy.

#### 3.3.6. High-Speed and Aggressive Maneurvering

High-speed and aggressive flight, where quick reaction times are essential, may be the most notable use of event cameras in UAVs. To perform aggressive flight maneuvers, such as making sharp turns, avoiding swift objects, and navigating through crowded areas, UAVs have been equipped with event cameras. A bio-mimetic fused vision system for microsecond-level target localization was created by [85], allowing UAVs to chase nimble targets and execute evasive maneuvers. Their edge-optimized solution supported high-speed control with low power consumption by combining event data and spiking neural models.

These systems could be used for drone racing, military evasion, or agile urban delivery, but their transfer from lab to field still requires generalizability. 

Table 8 summarizes the review of event camera vision system applications in UAVs.

### 3.4. Datasets and Open-Source Tools

In this section, we delve into the different datasets that are available for UAV applications using event cameras. The objective is to expose researchers to a wide array of event datasets and their challenges that are available specifically for UAV applications. Furthermore, we discussed various open-source tools and simulators, including their variation and challenges.

#### 3.4.1. Available Datasets for Event Cameras in UAVs Applications

Event camera vision systems are being used more often in UAVs for jobs requiring high temporal resolution, low latency, and effective data processing. These cameras function by detecting changes in the visual scene instead of taking entire image frames. Specialized datasets are required to maximize the usage of event cameras in UAV applications due to their distinct capabilities.

A.Event Camera Dataset for High-Speed Robotic TasksThis dataset includes high-speed dynamic scenes that are relevant to UAV maneuvers, like fast-paced tracking and navigation tasks. It provides ground truth measurements from motion capture systems along with event data, which makes it useful for benchmarking high-speed perception algorithms in UAVs [29]. They indicated that there are two recent datasets that also utilize DAVISs: [100,101]. The first study is designed for comparing algorithms that estimate optical flow based on events [100]. This dataset includes both synthetic and real examples featuring pure rotational motion (three degrees of freedom) within simple scenes that have strong visual contrasts, and the ground truth information was obtained using an inertial measurement unit. However, the duration of the recording of this dataset is not sufficient for a reliable assessment of SLAM algorithm performance [102].B.Davis Drone Racing DatasetThis is the first drone racing dataset, and it contains synchronized inertia measuring units, standard camera images, event camera data, and precise ground truth poses recorded in indoor and outdoor environments [103]. The event camera used for this dataset is miniDAVIS346 with a special resolution of 346 × 260 pixels, which proved to be of better quality than the one used by [29], which is DAVIS240C, with a resolution of 240 by 180 pixels.C.Extreme Event Dataset (EED)This dataset was collected using the DAVIS246B bio-inspired sensor across two scenarios. It was mounted on a quadrotor and on handheld devices for non-rigid camera movement [1]. This is the first event camera dataset that is specifically designed for moving object detection and was used as a benchmark dataset by [104] in their segmentation method to split a scene into independent moving objects.D.Multi-Vehicle Stereo Event Camera Dataset (MVSEC)MVSEC provides event data captured in a diverse set of environments, including indoor and outdoor scenes. It includes stereo event cameras mounted on a UAV, synchronized with other sensors like IMUs and standard cameras. The dataset is crucial for stereo depth estimation, visual odometry, and SLAM (Simultaneous Localization and Mapping) in UAVs [105]. This dataset was combined with the accuracy of the frame-based camera for high-speed optical flow estimation for UAV navigation with a validation of 19% error degradation sped up by 4x [98].E.RPG Group Zurich Event Camera DatasetThe research team at the University of Zurich is the leading force in advancing research on event-based cameras. These datasets were generated from their iniLabs using the DAVIS240C sensor. They were generated for different motions and scenes and contain events, images, IMU measurements, and camera calibrations. The output is available in text files and ROSbag binary files, which are compatible with Robot Operating System (ROS). This dataset is a standard for the development and assessment of algorithms in pose estimation, visual odometry [89], and SLAM [39], especially within UAV applications, but its dataset scenarios may not cover real-world UAV environments, potentially constraining generalizability [29].F.EVDodgeNet DatasetThis dataset called the Moving Object Dataset (MOD) was created using synthetic scenes for generating “unlimited” amount of training data with one or more dynamic objects in the scene [23]. This is the first dataset to focus on event-based obstacle avoidance and was specifically generated for neural network training.G.Event-Based Vision Dataset (EV-IMO)The most well-known dataset created especially for event cameras integrated into UAV systems is the Event-Based Vision Dataset (EV-IMO). It has dynamic scenes with a variety of moving objects that mimic UAV flight situations. According to [51], this dataset is especially helpful for problems involving object tracking, motion prediction, and feature extraction from event-based data.H.DSECThis dataset is similar to MVSEC since it was obtained from a monochrome camera and LIDAR sensor for ground truth. However, the data from these two Prophese Gen 3.1 sensor event cameras has a resolution that is three times higher than from MVSEC [106].I.EVIMO2This dataset expanded on EV-IMO with improved temporal synchronization between sensors and enhanced depth ground truth accuracy. Using Prophesee Gen3 cameras (640 × 480 pixels), it supported more complex perception tasks including optical flow and structure from motion [107].

#### 3.4.2. Simulators and Emulators

The development and testing of UAVs with event cameras prior to their deployment in real-world situations requires the use of simulators and emulators. Developers can test algorithms in a controlled setting by using tools such as the event camera simulator (ESIM), which offers an online environment. According to [108] these technologies utilize simulated scenarios to mimic the output of event cameras, which enables software developers to improve their product without requiring on-site testing. Additionally, AirSim has also been used by [46,53,68]. However, using Microsoft AirSim and Unreal Engine introduced significant computational overhead, severely limiting the number of training iterations [109]. A state-of-the-art simulator for converting video frames to realistic DVS event camera was used by [60] to generate synthetic event dataset for infrastructure defects.

Furthermore, XTDrone served as a simulation environment for testing dynamic obstacle avoidance [62].

ARobotic Operating System (ROS)When creating UAVs equipped with event cameras, the Robotic Operating System (ROS) is frequently utilized. ROS offers an adaptable structure for combining sensors, handling information, and managing unmanned aerial vehicles. Event cameras require an event-driven architecture, which is supported with packages that make real-time processing and data fusion easier. Because ROS provides a wide range of libraries and tools for managing sensor data, path planning, and control algorithms, it is very beneficial. Rapid prototyping and testing are made possible by the collaborative development environment that ROS’s open-source nature supports [110].BGazebo and RvizThe popular simulation and visualization tools Gazebo and Rviz are utilized with ROS for UAV development. With Gazebo, UAVs may be tested in virtual environments with dynamic objects and changing lighting, an essential feature for event cameras. Gazebo is a 3D simulation environment. Rviz, on the other hand, makes it simpler to debug and improve algorithms by providing real-time visualization of sensor data and the UAV’s condition as it was used by [88]. In Table 9 is the list of open-source event camera simulators and source codes.

Figure 5 highlights the top 10 institutions in the research dataset. ETH Zürich leads the group, closely followed by Universität Zürich and Universidad de Sevilla. The Institute of Neuroinformatics also makes a significant contribution, alongside CNRS Centre National de la Recherche and the National University of Defense Technology. Additional key contributors include Beihang University, Tsinghua University, Delft University of Technology, and the Air Force Research Laboratory. The distribution shows a strong concentration of research activity among prominent European and Asian technical universities, with Swiss institutions notably prominent. The involvement of defense-related organizations indicates that the research area likely has military or security applications.

In Table 10, we summarized the most common event camera used by the year. From 2015 to 2017, UAV obstacle avoidance relied mainly on low resolution DVS128/DVS for indoor navigation. By 2019–2020, more diverse sensors like SEES1 and DAVIS240C were introduced for real-world tests. In 2022–2023, the DAVIS family and Celex4 gained popularity due to their higher resolution supporting hybrid frame-event sensing. From 2024 until the present date, higher-resolution, domain-specific sensors such as CeleX-5 and Prophesee EVK4-HD emerged for specialized tasks.

Figure 6 illustrates the evolution of research methods in UAV event camera studies from 2015 to 2025. Model-based methods have been the dominant approach throughout the period, showing consistent growth demonstrating their continued relevance as a foundational technique. Hybrid or fusion sensor approaches first appeared in 2018 and have experienced significant growth, particularly from 2020 onwards, indicating increasing interest in combining multiple methodologies and sensor fusion techniques. Learning-based methods emerged in 2019 and have shown substantial expansion, with notable acceleration from 2021 through 2024, reflecting the rising adoption of deep learning, reinforcement learning, and Transformer-based architectures. Neuromorphic techniques have been employed intermittently since 2019, with relatively modest but consistent representation across recent years, demonstrating ongoing interest in bio-inspired computing and spiking neural networks for UAV applications. This trend reveals a shift from exclusively model-based approaches in early years toward a diverse methodological ecosystem, with all four method categories actively employed by 2024–2025, suggesting that the field has matured into a multi-faceted research domain that integrates traditional geometric principles with modern machine learning and bio-inspired techniques.

## 4. Discussion

### 4.1. Replacing Frame-Based Camera with Event Camera in UAV Applications

Ref. [112] demonstrated on an ornithopter robot how frame-based cameras performed very well with corner detection, but event camera excelled in dynamic range and robustness to motion blur. Despite similar performance in some tasks, event cameras consumed less power, though they faced processing bottlenecks with high event rates (~0.97 million events/s). And under harsh illumination conditions, event-based cameras outperformed framed-based cameras in UAV object tracking with improved image enhancement and up to 39.3% higher tracking accuracy [79]. This was also proved by [113] with fault tolerance in autonomous quadrotor flight despite rotor failure. Event-based cameras are replacing frame-based cameras in high-speed operation, high dynamic range, and low light or harsh illumination in UAV applications due to the robustness of the cameras; however, frame-based cameras still perform better especially in a static environment.

### 4.2. Challenges in Software Development and Deployment for Event Camera Vision Systems in UAVs

The asynchronous and sparse nature of the data creates special issues for developing software for event cameras in UAVs. To handle event-based data, traditional vision algorithms which are frequently frame-based need to be modified or completely redesigned. Development may also be hampered by the lack of uniformity in event camera data formats and processing technologies [22]. Creating software that works is made more difficult by the requirement for specific understanding in both robotics and computer vision. Computational load is another issue that needs to be addressed by developers because real-time processing of high-frequency event streams demands a lot of processing power and effective code optimization [21]. To fully utilize event cameras in UAVs and enable improved capabilities in dynamic and unpredictable environments, certain hardware and software components are necessary [21].

Event noise during fast UAV maneuvers: Event cameras are sensitive to intensity changes. This sensitivity can lead to noisy activations that necessitate sophisticated filtering, especially in cluttered, multi-object scenes [114].

Sensor–IMU synchronization challenges: Robust navigation and perception often require fusing event cameras with other sensors like Inertial Measurement Units (IMUs). Systems such as [39,88] combine event data with inertial measurements. However, these techniques still require significant sensor fusion to ensure stability over extended missions, implicitly highlighting the need for precise synchronization.

### 4.3. Evolution of Event Camera Dataset for UAV Applications

The evolution of event camera datasets for UAV applications has progressed through three distinct generations since 2017, showing remarkable technical advancement. Starting with foundational collections using low-resolution DAVIS (240 × 180 pixels), these datasets have evolved to incorporate high-resolution Prophesee cameras (up to 1280 × 720 pixels), sophisticated ground truth methodologies, and diverse environmental settings. The MVSEC dataset [105] has emerged as the most widely adopted benchmark due to its comprehensive multi-vehicle scenarios and stereo vision capabilities, with over 500 citations in the literature. For researchers focusing on high-speed drone applications, the UZH-FPV Drone Racing dataset [103] offers superior sub-millisecond precision essential for racing applications, while those requiring detailed motion segmentation should utilize EV-IMO [51] with its pixel-wise ground truth. The DSEC dataset [106] provides the best option for high-resolution perception tasks, whereas EVIMO2 [107] represents the current state of the art for researchers requiring advanced sensor fusion and depth estimation capabilities.

Despite significant progress, current event camera datasets for UAVs face substantial limitations that impede broader adoption and real-world application. These challenges include restricted operational scenarios predominantly in controlled environments rather than authentic UAV missions; application bias toward racing and obstacle avoidance with insufficient representation of inspection, mapping, or multi-UAV operations; and persistent technical issues including inconsistent calibration approaches, non-standardized data formats, and varying annotation quality. For specific applications, researchers should select datasets strategically: obstacle avoidance systems should build upon EVDodgeNet; autonomous racing should leverage UZH-FPV; SLAM applications are best served by MVSEC’s diverse environments, while low-light operations benefit most from the EED’s unique strobe light scenarios. Future datasets must address the critical gap in long-duration autonomous flights, adverse weather conditions, and multi-UAV interaction scenarios to facilitate the transition from laboratory research to commercial applications in inspection, delivery, and surveillance domains.

### 4.4. Comparing the Algorithm

Table 11 presents a comparative summary of the major algorithmic paradigms used in this review. It highlights key trade-offs in all the approaches across latency, accuracy, robustness, and energy consumption, illustrating how geometry-based, learning-based, neuromorphic, and hybrid approaches address fast and dynamic autonomous aerial flights scenarios with varying performance, computational demands, and practical deployment constraints.

## 5. Conclusions

This paper has presented a comprehensive review of research on the integration of event camera vision systems on unmanned aerial vehicles (UAVs) from 2015 to 2025. The review emphasizes how event-based vision can revolutionize the performance of UAVs, especially in areas such as dynamic obstacle avoidance, high-speed navigation, HDR environments, and GPS-denied localization where traditional frame-based cameras have significant limitations. The review highlights the increasing depth and breadth of work in this interdisciplinary subject by thematically organizing the literature into datasets, simulation tools, algorithmic approaches (neuromorphic, learning-based, geometric, and hybrid fusion), and application domains. Even though there has been much research on this camera in robotics, event camera vision systems are still not widely used in real-life UAV applications. The absence of established evaluation methodologies, inadequate real-world validation, immature simulation platforms, hardware integration limitations, and inadequate datasets with ground truth are some of the main obstacles. These restrictions show a gap between practical needs for reliable, real-time UAV operation and exciting academic research.

Despite many challenges, this review has shown that event camera vision systems hold immense potential in advancing UAV autonomy, particularly in real-life complex and dynamic environments where conventional frame-based cameras fall short.

## Figures and Tables

**Figure 1 sensors-26-00081-f001:**
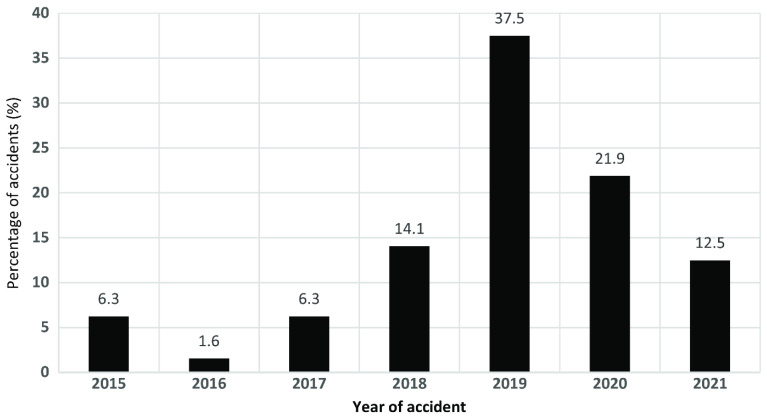
Reported drone or UAV accidents by year of occurrence [12].

**Figure 2 sensors-26-00081-f002:**
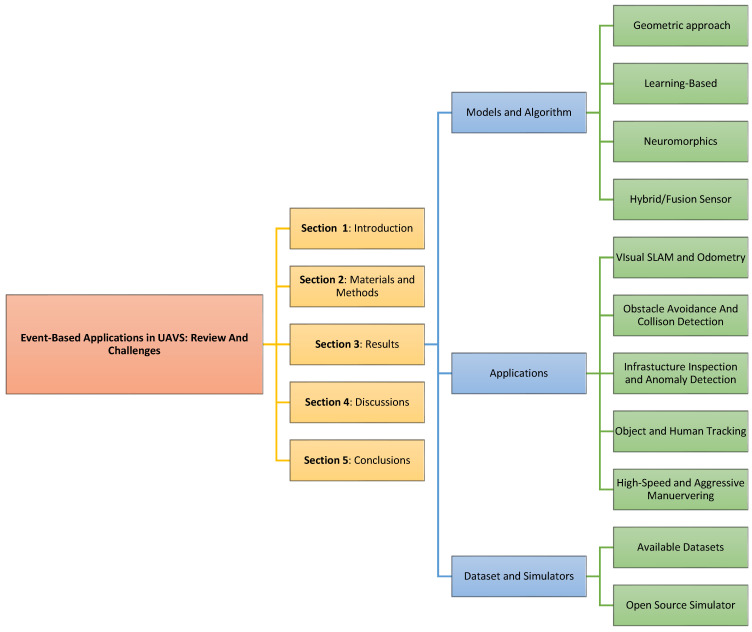
A summary of organization of research.

**Figure 3 sensors-26-00081-f003:**
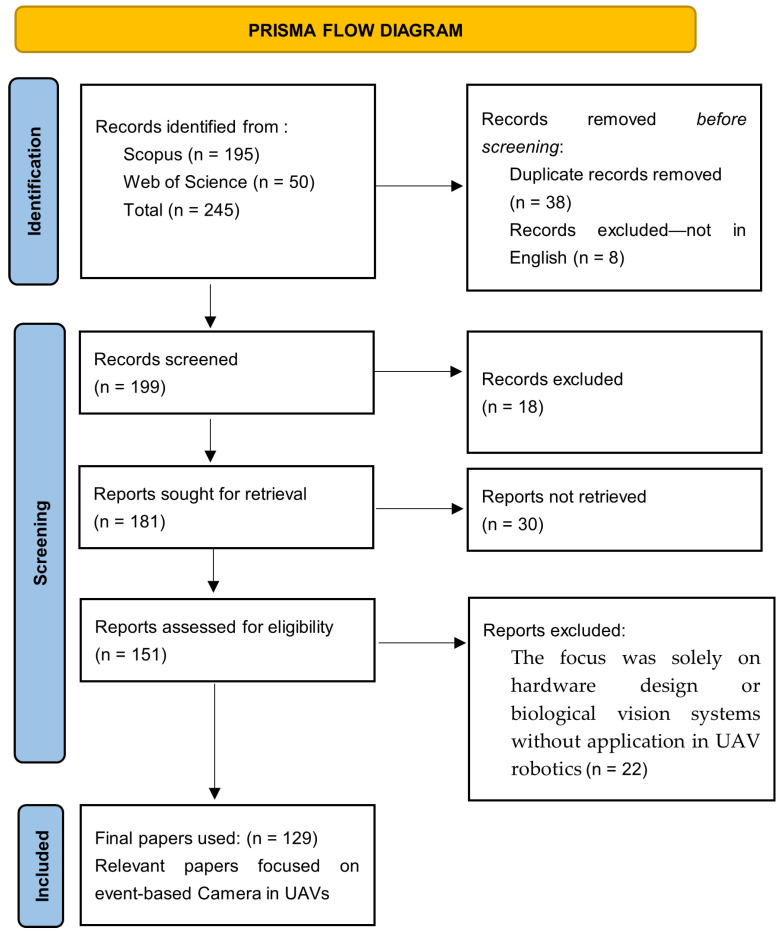
PRISMA flowchart.

**Figure 4 sensors-26-00081-f004:**
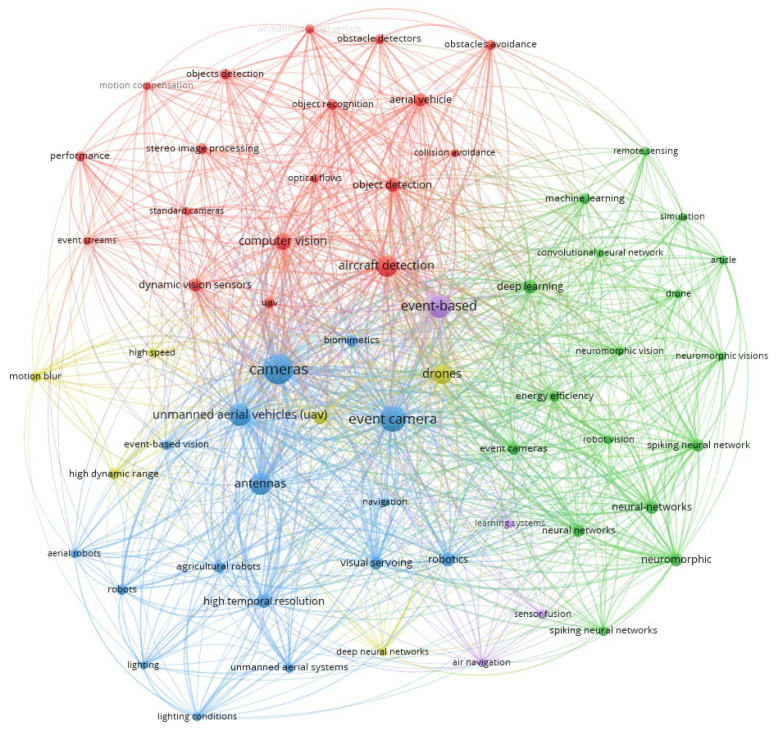
Bibliometric mapping of event camera applications in UAVs.

**Figure 5 sensors-26-00081-f005:**
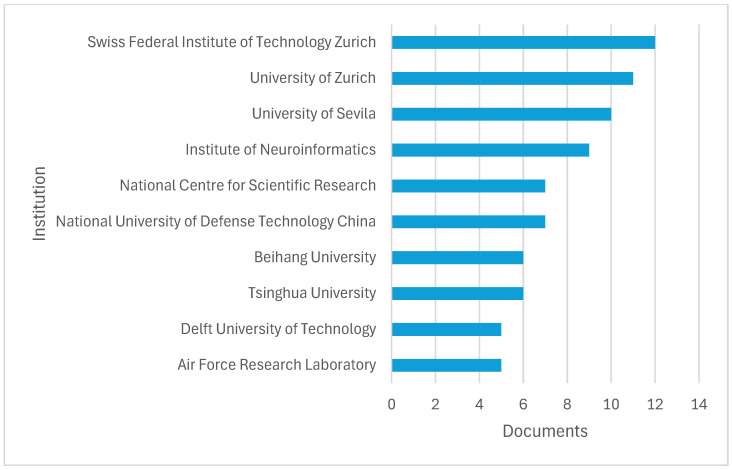
Top global research institutions publishing on event-based vision and UAV technologies.

**Figure 6 sensors-26-00081-f006:**
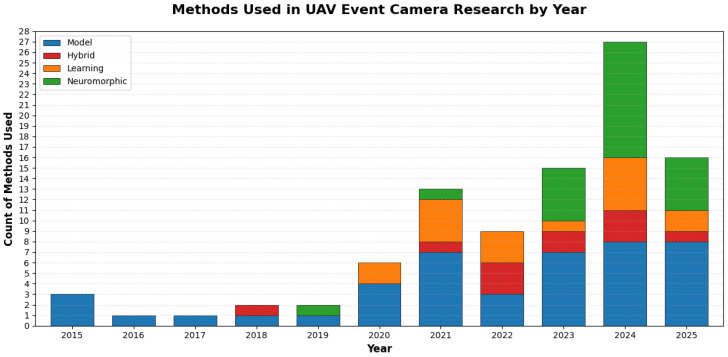
Summary of the method used over the year.

**Table 1 sensors-26-00081-t001:** Types of event cameras.

Type	Operation	Gaps
Dynamic Vision Sensors (DVSs) [28]	Detecting variations in brightness is the sole method used by DVS, the most popular kind of event camera. When the amount of light in the scene varies enough, each pixel in a DVS independently scans the area and initiates an event. With their high temporal resolution and lack of motion blur, DVSs work especially well in situations involving rapid movement. The DVS has a number of benefits over conventional high-speed cameras, one of which being their incredibly low data rate, which qualifies them for real-time applications.	Despite these capabilities, integrating DVSs with UAVs remains a challenge, especially on the issues of real-time processing and data synchronization [21]. A lack of standardized datasets is also making it difficult to evaluate the performance of DVS camera-based UAV applications [29]
Asynchronous Time-based Image Sensors (ATIS) [30]	ATIS combines the capability of capturing absolute intensity levels with event detection. Not only can ATIS record events that are prompted by brightness variations, but it can also record the scene’s actual brightness at times. Rebuilding intensity images alongside event data is made possible by this hybrid technique, which enables greater information acquisition and is especially helpful for applications that need both temporal precision and intensity information.	Data from an event-based ATIS camera can be noisy, especially in low-light conditions. So, there is a need for an efficient noise filtering model to address this [31]
Dynamic and Active Pixel Vision Sensor (DAVIS) [2]	DAVISs combine traditional active pixel sensors (APS) and DVS capability. Because of its dual-mode functionality, DAVISs may be used as an event-based sensor to identify changes in brightness or as a conventional camera to record full intensity frames. The DAVIS’s dual-mode capacity makes it adaptable to a variety of scenarios, including those in which high-speed motion must be monitored while retaining the ability to capture periodic full-frame photos.	This capability of combining both APS and DVS capability poses challenges in complex data integration and sensor fusion [32]
Color Event Cameras [33]	Color event cameras are one of the more recent innovations that increase the functionality of traditional DVSs by capturing color information. These sensors enable the camera to record color events asynchronously by detecting changes in intensity across various color channels using a modified pixel architecture. This breakthrough enables event cameras to be utilized in more complicated visual settings where color separation is critical.	There is a scarcity of comprehensive dataset repositories specifically for training and evaluating models that use this camera [34]

**Table 2 sensors-26-00081-t002:** Research paper selection process.

keywords	event based camera, event-based camera, dynamic vision sensor, dvs, unmanned aerial vehicle, uavs, drone
Databases	Scopus and Web of Science
Boolean operator	OR, AND
Language	English
Year of publication	2015 to 2025
Inclusion criteria	English, peer review journal articles and conferences proceeding, addressed the use of event cameras on DVSs in the context of UAVs
Exclusion criteria	Prior to 2015, not English, duplicate, focus was solely on hardware design or biological system without application in UAV
Document type	Published scientific papers in academic journals, conference papers

**Table 3 sensors-26-00081-t003:** Geometric approach equation annotation.

Symbol	Description	Units/Notes
I	Pixel intensity	Arbitrary units
(x, y)	Pixel coordinates in 2D image plane	Pixels
∆ log(I)	Change in logarithmic intensity	Unitless
∇ log(I)	Spatial gradient of the logarithmic intensity	Unitless
$\dot{u}$ (dot u)	Motion field (spatial velocity of pixels)	Pixels per unit time
∆t	Time interval	Seconds
C	Contrast threshold for triggering events	Threshold value
e_k_ = (x_k_, t_k_, p_k_)	Event tuple: spatial coordinate, timestamp, polarity	x_k_ in pixels; t_k_ in seconds; p_k_ ∈ {+1,−1} polarity indicator
δ (delta function)	Approximation of Dirac delta by Gaussian	Unitless, probability density function
Ω (Omega)	Image domain	Region within pixels
$\mu_{I}$	Mean intensity over the image domain	Intensity units
θ (theta)	Parameters of motion or warp function	Typically includes angles and translations

**Table 4 sensors-26-00081-t004:** Summary of relevant contributions using Geometric approaches.

Author	Year	DVS Type	Evaluation	Application/Domain	Model	Future Direction
[43]	2015	DVS 128	Real indoor test flight with a miniature quadcopter	Navigation	Flexible algorithm infers motion from adjacent pixel time differences	The study targeted an indoor environment. Dynamic scenes with more complex environments are required.
[38]	2019	DVS 128	Dataset from [29]	Vision aid landing	Adaptive block matching optical flow	Further work should focus on the robustness and the accuracy of landmark detection especially in a complex scene.
[22]	2020	SEES1	Real-world experiment with quadrotor	Dynamic obstacle avoidance	(IMU)’s angular velocity average for ego-motion, DBSCAN for clustering and APF for obstacle avoidance	This approach model obstacles as ellipsoids and relies on sparse representation. Extending this approach to more complex environments with non-ellipsoidal obstacles and cluttered urban environments remains a challenge.
[44]	2023	Simulated DVS with resolution of 1024 × 768	Simulator based on OSG-Earth	Navigation and control	Mutual information for image alignment	The algorithm is limited to 3-DoF displacement (translation) and does not incorporate changes in orientation, limiting its capability to fully determine the 6-DoF pose.
[45]	2023	DAVIC 240C	MVSCEC Dataset	Navigation		The author recommended a complete SLAM framework for high-speed UAV based on even camera
[46]	2024	CeleX-5	Real world with UAV and simulation with Unreal Engine and AirSim	Powerline inspection and tracking	The EAPTON (Event-based Antinoise Powerlines Tracking with ON/OFF Enhancement)	Lack of dataset in that domain and inability of the model to accurately distinguish between power lines and nonlinear object in a complex scene.
[47]	2024	DAVIS 326	Real-world with octorotor UAV indoors and outdoors	Load transport; cable swing minimization	Point cloud representation and a Bézier curve combined with Nonlinear Model Predictive Controller	Future work could focus on enhancing event detection robustness during larger cable swings, developing more sophisticated fusion techniques, and extending the method’s applicability to dynamic, highly noisy environments.

**Table 5 sensors-26-00081-t005:** Summary of relevant contributions using learning-based approaches.

Author	Year	Event Camera Type	Method of Evaluation	Application/Domain	Model	Future Direction
[53]	2022	DVS	The system was evaluated via simulation trials in Microsoft AirSim.	Event-based object detection, obstacle avoidance	Deep reinforcement learning	The study highlights the need to optimize network size for better perception range, design new reward functions for dynamic obstacles, and incorporate LSTM for improved dynamic obstacle sensing and avoidance in UAVs
[54]	2022	Prophese 640 by 480	Real-world on small UAV	Localization and Tracking	YOLOv5 and k-dimensional tree	The research primarily focused on 2D tracking and future work should be extended to 3D tracking/control
[55]	2023	DAVIS346	Real-world testing with a hexarotor UAV installed with both event- and frame-based cameras and simulation in Matlab Simulink	Visual servoing robustness	Deep reinforcement learning	The proposed DNN with noise protected MRFT lacks robust high-speed target tracking under noisy visual sensor data and slow update-rate sensors; future directions include developing adaptive system identification for high-velocity targets and optimizing neural network-based tuning to improve real-time accuracy under varying sensor delays and noise conditions
[56]	2024	Prophesee camera EVK4–HD	To bridge the data gap, the first large-scale high-resolution event-based tracking dataset called EventVot was produced through UAVs and used for real-world evaluation	Obstacle localization; navigation	Transformer-based neural networks	The high-resolution capability of the Prophesee EVK4–HD camera (1280 × 720) opens new avenues for improving event-based tracking, but it also introduces additional challenges, such as increased computational complexity and data processing requirements
[57]	2024	DAVIS 346c	Real-world testing in a controlled environment with hexacopter	Obstacle avoidance	Graph Transformer Neural Network (GTNN)	Real-world experiment in a complex environment is limited in the literature
[58]	2024	n.a	Real-world experiment with s500 quadrotor	Crack detection/inspection	Unet and YOLOv8n-seg network	Explore the use of actual event camera sensor to directly capture real temporal information
[59]	2025	DVS 128	Evaluation was done using N-MNIST, N-CARS, CIFAR10-DVS dataset	Object detection	A motion-aware branch (MAB) enhances 2D CNNs	future research could focus on optimizing the input patches by filtering out meaningless or noisy patches before they are fed into MAB

**Table 6 sensors-26-00081-t006:** Summary of relevant contributions of neuromorphic computing approaches.

Author	Year	Event Camera Type	Method of Evaluation	Application/Domain	Model	Future Direction
[64]	2017	DVS	Real-world recorded data from a DVS mounted on a QUAV	Obstacle avoidance	Spiking neural network model of LGMD	Integrate motion direction detection (EMD) and enhance sensitivity to diverse stimuli
[65]	2019	DVS240	Real-world testing in indoor environment using the actual data from a DVS and simulation testing using data that was processed through an event simulator (PIX2NVS)	Drone detection	SNNs trained using spike-timing-dependent plasticity (STDP).	The model was tested in an indoor environment. Exploring the system in a resource-constrained environment is critical
[66]	2020	DAVIS240C	Real-world experiment on two-motor 1-DOF UAV	SLAM	PID+SNN	The authors suggested the potential for integrating adaptation mechanism and online learning into the SNN-based controllers by utilizing the chip’s on-chip plasticity
[67]	2021	DAVIS 240C	Real-world experiment on dualcopter	Autonomous UAV control	Hough transform with PD controller	Full on-chip neuromorphic integration for direct communication with flight controllers to reduce latencies and delays
[68]	2023	n.a	Simulated DVS implemented through the v2e tool within an AirSim environment	Obstacle avoidance	Deep Double Q-Network (D2QN) integrated with SNN and CNN	Improve network architecture for better performance in real world
[60]	2023	DAVIS 346	Validated with simulated and collected datasets	Civil infrastructure inspection	DNN and SNN	Creating a real event-camera-based dataset for extreme illumination effects and testing SNNs on a real embedded neuromorphic device
[63]	2024	DVS 240	Real drone	Autonomous UAV control	SNN and ANN	The author suggested the best approach to have an energy-efficient system is to make the entire drone system neuromorphic
[69]	2024	n.a	Simulation with Gazebo and ROS	Autonomous Control	SNN and ANN	A real-world simulation is suggested
[70]	2024	DVS	Real-world experiment with drone	People Detection	SNN STDP	The author suggested multi-person detection and implementation of neuromorphic chip for low power, low latency
[71]	2024	Prophese EVK4 HD	Neuromorphic MNIST (N-MNIST) dataset	Motion Estimation	Fuzzy SNN	Collecting actual event data in a mock Mars environment
[61]	2024	n.a	DDD17 dataset	Navigation	SNN with Surrogate Gradient Learning	Implementing the model on low-power neuromorphic hardware
[72]	2024	DVXplorer Mini camera	Simulation on neurorobotics platform	Asset Monitoring	SNN and Kalman filtering	Further research includes the port of optical flow computation to neuromorphic hardware and the full port of the system onto a real drone, for real-world assessment
[62]	2024	DVS	Simulation was performed in XTDrone-based	Obstacle avoidance	Dynamic neural field with Kalman filter	Deploy the lightweight SNN onto neuromorphic hardware for obstacle detection
[73]	2024	ESIM to generate the event	Synthetic data from ESIM	Obstacle avoidance	LGMD with FSN	Deploying the model in a complex scene
[74]	2025	n.a	Real-world experiment on drone	Obstacle avoidance	CEF and LEM	Extending the design principle beyond obstacle avoidance to navigation

**Table 7 sensors-26-00081-t007:** Summary of relevant contributions of hybrid approaches.

Author	Year	Event Camera Type	Method of Evaluation	Application/Domain	Model	Future Direction
[39]	2018	DVS	The result was evaluated with [29] dataset	SLAM	Hybrid state estimation combining data from event and standard cameras and IMU.	Future work should expand this multimodal sensor to more complex real-world applications
[76]	2021	DVX Explorer 640 by 480	Real-world quadrotor	Object detection and avoidance	Fuses IMU/depth.	Integrating avoidance algorithms based on motion planning, which would consider static and dynamic scenes
[75]	2022	DAVIS 346	6DOF quadrotor, using dataset from [40]	VIO	VIO model combining event camera, IMU, and depth camera for range observations.	According to the author, the effect of noise and illumination on the algorithm is worth studying in the next step
[77]	2023	DAVIS346	Real-world in a static and dynamic environment using AMOV-P450 drone	Motion tracking and obstacle detection	It fuses asynchronous event streams and standard image utilizing nonlinear optimization through Photometric Bundle Adjustment with sliding windows of keyframes, refining pose estimates.	Future work aims to incorporate edge computing to accelerate processing
[78]	2024	Prophesee EVK4-HD sensor	Two insulator defect datasets, CPLID and SFID	Power line inspection	YOLOv8.	While the experiment used reproduced event data derived from RGB images, the authors note that real-time captured event data could better exploit the advantages of neuromorphic vision sensors
[79]	2024	n.a	Simulated data and real-world nighttime traffic scenes captured by a paired RGB and event camera setup on drones	Object Tracking	Dual-input 3D CNN with self-attention.	Integration of complementary sensors such as LIDAR and IMUs for depth-aware 3D representations and more robust object tracking
[80]	2024		Real-world testing on quadrotor both indoors and outdoors	VIO	PL-EVIO which tightly coupled optimization-based monocular event and inertial fusion.	Extending the work to event-based multi-sensor fusion beyond visual-inertial, such as integrating LiDAR for local perception and visible light positioning or GPS for global perception, to further exploit complementary sensor advantages

**Table 8 sensors-26-00081-t008:** Summary of review of the applications of event camera vision systems in UAVs.

Cited Works	Application Area	Challenges/Future Directions
[29,39,41,43,55,75,80,83,86,87,88,89,90]	Visual SLAM and Odometry	Performance degrades in low-texture or highly dynamic scenes; need for stronger sensor fusion (e.g., with IMU, depth); robustness under aggressive maneuvers.
[22,23,53,57,62,64,68,73,74,77,91,92,93,94,95,96]	Obstacle Avoidance and Collision Detection	Filtering noisy activations; setting adaptive thresholds in cluttered, multi-object environments; scaling to dense urban or swarming scenarios.
[80,81]	GPS-Denied Navigation and Terrain Relative Flight	Requires fusion with depth and inertial data for stability; limited long-term robustness; neuromorphic SLAM hardware still in early stages.
[32,58,60,78,82]	Infrastructure Inspection and Anomaly Detection	Lack of large, annotated datasets; absence of benchmarking standards; need for generalization across varied materials and lighting.
[1,27,45,46,52,54,56,70,76,79,84,97]	Object and Human Tracking in Dynamic Scenes	Sparse, non-textured data limits fine-grained classification; re-identification with event-only streams remains difficult; improved multimodal fusion needed.
[79,85,98,99]	High-Speed and Aggressive Maneuvering	Algorithms need to generalize from lab to real world; neuromorphic hardware maturity; power-efficiency vs. control accuracy trade-offs.

**Table 9 sensors-26-00081-t009:** Open-source event camera simulator and source codes.

S/N	Name	Inventor	Year	Source
1	ESIM (Event Camera Simulator)	[108]	2018	https://github.com/uzh-rpg/rpg_esim (accessed on 5 October 2025)
2	ESVO (Event-Based Stereo Visual Odometry)	[90]	2022	https://github.com/HKUST-Aerial-Robotics/ESVO (accessed on 5 October 2025)
3	UltimateSLAM	[40]	2018	https://github.com/uzh-rpg/rpg_ultimate_slam_open (accessed on 5 October 2025)
4	DVS ROS (Dynamic Vision Sensor ROS Package)	[99]	2015	https://github.com/uzh-rpg/rpg_dvs_ros (accessed on 5 October 2025)
5	rpg_evo (Event-Based Visual Odometry)	[111]	2020	https://github.com/uzh-rpg/rpg_dvs_evo_open (accessed on 5 October 2025)

**Table 10 sensors-26-00081-t010:** Summary of the types of event cameras used over the years.

Year	Event Camera Type(s)
2015	DVS 128
2017	DVS
2018	DVS
2019	DVS 128, DVS 240
2020	SEES1, DAVIS 240C
2022	Celex4 Dynamic Vision Sensor, DAVIS 346
2023	DAVIS, DAVIS 240C, DAVIS 346, DAVIS 346c
2024	CeleX-5, Prophesee EVK4-HD, DAVIS 326
2025	DVS346

**Table 11 sensors-26-00081-t011:** Comparing the robustness, latency, accuracy, and energy consumption of different algorithms.

Algorithmic Category	Latency	Accuracy	Robustness	Energy Consumption	Notes/Limitations
Geometry approach	Very low (microsecond-level)	Moderate to high in controlled/simple environments	Sensitive to noise, scene sparsity, and dynamic elements	Moderate, suitable for embedded systems	Mathematical rigor with optical flow, but limited in complex scenes and textureless environments
Learning-Based	Moderate, varies with model complexity	Generally high, can outperform model-based in complex tasks	Improved adaptability to complex and dynamic environments	High, due to training and inference overhead on DNN/GTNN models	Needs large labeled datasets with ground truth real-world validation limited
Neuromorphic	Ultra-low latency due to spike-based processing	Competitive, especially in reactive tasks	High robustness to motion blur and high dynamic range scenes	Very low power, hardware-accelerated (e.g., Intel Loihi)	Hardware scarcity and immature platforms restrict broad adoption
Hybrid/Fusion	Variable, depends on sensor fusion algorithms	Potentially highest due to multi-source data fusion	Enhanced robustness combining strengths of multiple sensors	Moderate to high, depending on system complexity	Integration challenges; immature simulation platforms and datasets

## Data Availability

No new data were created or analyzed in this study.

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
