# Peer review of "Event-Based Vision Application on Autonomous Unmanned Aerial Vehicle: A Systematic Review of Prospects and Challenges"

_sensors, 2025, doi:10.3390/s26010081_

Round 1
Reviewer 1 Report
Comments and Suggestions for Authors
Authors conducted UAV-focused systematic review of event-based cameras, analyzing articles collected through a PRISMA procedure from 2015–2025. They proposed structured assessment of algorithms, datasets, simulators, and application areas, organizing existing work into geometric, learning-based, neuromorphic, and hybrid sensor-fusion approaches. Prepared review identifies that event cameras offer clear advantages in latency, motion-blur robustness, and high-speed perception, but practical UAV adoption remains limited due to insufficient datasets, lack of standard evaluation metrics and hardware-integration challenges.
Manuscript includes several charts summarizing publication trends, method usage, and bibliometric patterns but lacks quantitative performance comparisons across studies, such as cross-study accuracy, latency, or robustness metrics. Review describes existing approaches but does not critically evaluate how these methods perform under real UAV operating conditions. Hardware-integration challenges and the fidelity of event-camera simulators are only briefly addressed leaving key implementation aspects underexplored.
Authors shoudl improve manuscript considering:
a)adding compact comparison matrix summarizing key algorithmic performance indicators such as latency, accuracy, robustness and energy consumption
b)expanding discussion of practical deployment barriers, including event noise during fast UAV maneuvers, sensor–IMU synchronization challenges, calibration difficulties, and vibration-induced artifacts.
c)providing more detailed comparison of available simulators paying attention to ESIM fidelity, AirSim event-camera extensions and integration challenges in ROS/Gazebo environments
d)improving the description of the SLR methodology by removing redundancies and clarifying screening steps, inclusion/exclusion criteria and the overall methodological flow
It is required that authors should also mention about sensor limitations under varying environmental conditions in unmanned systems (https://doi.org/10.1109/ME61309.2024.10789716) as well as consider operational reliability of event-based cameras in aerial applications (https://doi.org/10.1109/DSN-W65791.2025.00066) along with other relevant studies to view of real-world constraints affecting sensor performance.
Author Response
Comments a: adding compact comparison matrix summarizing key algorithmic performance indicators such as latency, accuracy, robustness and energy consumption
Response a: Thank you for pointing this out. We agreed with this comment, and a comparison matrix has been included in the revision on page 34
Comments b: expanding discussion of practical deployment barriers, including event noise during fast UAV maneuvers, sensor–IMU synchronization challenges, calibration difficulties, and vibration-induced artifacts.
Response b: Thank you for pointing this out. We agreed with this comment and it has been addressed on page 8 of the revised article
Comments c: providing more detailed comparison of available simulators paying attention to ESIM fidelity, AirSim event-camera extensions and integration challenges in ROS/Gazebo environments.
Response c: Thank you for point this out, comparison of the available simulator and integration challenges with the ROS/Gazebo is very important. This has been addressed on page 28-29 of the revised article
Comments d: improving the description of the SLR methodology by removing redundancies and clarifying screening steps, inclusion/exclusion criteria and the overall methodological flow
Response: Thank you for pointing this out. A proper clarification of the inclusion and exclusion criteria has been indicated in the page 8 on revised article
Also, the article (https://doi.org/10.1109/DSN-W65791.2025.00066) you suggested is very insightful as it pointed out the need for event camera which helps with motion blur and occlusion caused by weather effect for autonomous navigation in a UAV-based deforestation system.
Reviewer 2 Report
Comments and Suggestions for Authors
Dear Authors,
- The paper provides a broad and timely systematic review of event-camera-based UAV perception, covering background, algorithms (geometric, learning-based, neuromorphic, and hybrid), applications, datasets, and tools. However, the contribution would be clearer if the authors explicitly contrasted this review with prior surveys on event cameras and UAVs, highlighting what is new.
- Overall, the writing is clear, but there are occasional long sentences and minor grammatical issues (e.g., spacing, pluralization, missing articles). A careful language edit would improve readability, especially in the Introduction and Background sections.
- Some background text on UAV applications and event camera basics is quite extensive and partially repeated later. The paper could be tightened by shortening the generic UAV background and focusing more on event-based UAV-specific insights.
- Figures such as the PRISMA flowchart, research structure, and bibliometric maps are informative, but some captions could be more self-contained (briefly explaining what conclusions the reader should draw). For tables summarizing contributions, consider consistent column naming and a clearer indication of what “future directions” imply for each work.
- The geometric section includes mathematical formulations (e.g., contrast maximization, IWE), but some variables are not defined immediately or consistently. A short notation table would help non-expert readers.
- The review gathers many works and provides useful summaries, but the level of critical comparison varies by section. For example, geometric approaches are analyzed with equations and limitations, whereas learning-based and neuromorphic approaches sometimes read more descriptive than comparative.
- The paper mentions several future directions (better datasets, standardized metrics, improved hardware integration, neuromorphic computing, and multi-UAV scenarios). These could be sharpened by formulating concrete research questions or design guidelines for the community (e.g., “What properties should the next-generation event-UAV dataset have?” “What metrics should become standard?”)
- Among the reviewed learning-based methods, which architectures (CNN, transformer, graph-based, or spiking) appear most promising for deployment on embedded UAV hardware, considering compute and power constraints?
- The list of references needs to be consistent and use appropriate formatting.
Author Response
Comments 1: The paper provides a broad and timely systematic review of event-camera-based UAV perception, covering background, algorithms (geometric, learning-based, neuromorphic, and hybrid), applications, datasets, and tools. However, the contribution would be clearer if the authors explicitly contrasted this review with prior surveys on event cameras and UAVs, highlighting what is new.
Response 1: Thank you for pointing this out. Although, to the best of our knowledge, there is no review or survey that specifically addressed the deployment of event cameras on UAV applications. However, we have provided similar reviews in respect of event cameras and autonomous vehicles, but they are not directly or dynamic vision sensor, and this has been added to section 1.4 on the revised article.
Comments 2: Overall, the writing is clear, but there are occasional long sentences and minor grammatical issues (e.g., spacing, pluralization, missing articles). A careful language edit would improve readability, especially in the Introduction and Background sections.
Response 2: Thank you for pointing this out. Readability has been enhanced in the introduction and background sections.
Comments 3: Some background text on UAV applications and event camera basics is quite extensive and partially repeated later. The paper could be tightened by shortening the generic UAV background and focusing more on event-based UAV-specific insights.
Response 3: Thank you for pointing this out. This has been addressed in the background section from pages 3-5.
Comments 4: Figures such as the PRISMA flowchart, research structure, and bibliometric maps are informative, but some captions could be more self-contained (briefly explaining what conclusions the reader should draw). For tables summarizing contributions, consider consistent column naming and a clearer indication of what “future directions” imply for each work.
Response 4: Thank you for pointing this put. This has been addressed.
Comments 5: The geometric section includes mathematical formulations (e.g., contrast maximization, IWE), but some variables are not defined immediately or consistently. A short notation table would help non-expert readers.
Response 5: Thank you for pointing this out. A table that includes the notations has been added to page 15.
Comment 6: The review gathers many works and provides useful summaries, but the level of critical comparison varies by section. For example, geometric approaches are analysed with equations and limitations, whereas learning-based and neuromorphic approaches sometimes read more descriptive than comparative
Response 6: Thank you for pointing this out. This has been addressed in section 3.13
Comment 8: Among the reviewed learning-based methods, which architectures (CNN, transformer, graph-based, or spiking) appear most promising for deployment on embedded UAV hardware, considering compute and power constraints?
Response 8: Thank you for pointing this out. This has been addressed in section 3.12
Comments 9: The list of references needs to be consistent and use appropriate formatting.
Response 9: Thank you for pointing this out. This is due to the reference manager used not automatically updated. This has been addressed
Round 2
Reviewer 2 Report
Comments and Suggestions for Authors
Dear Authors,
My concerns and suggestions have been addressed.